# GeNSeg-Net: A General Segmentation Framework for Any Nucleus in Immunohistochemistry Images

## ABSTRACT

Immunohistochemistry (IHC) plays a crucial role in understanding disease mechanisms, diagnosing pathology and guiding treatment decisions. The precise analysis heavily depends on accurate nucleus segmentation. However, segmentation is challenging due to significant inter- and intra-nucleus variability in morphology and distribution, stemming from inherent characteristics, imaging techniques, tissue differences and other factors. While current deep learning-based methods have shown promising results, their generalization performance is limited, inevitably requiring specific training data. To address the problem, we propose a novel **Ge**neral framework for **N**ucleus **Seg**mentation in IHC images (GeNSeg-Net). GeNSeg-Net effectively segments nuclei across diverse tissue types and imaging techniques with high variability using a small subset for training. It comprises an enhancement model and a segmentation model. Initially, all nuclei are enhanced to a uniform morphology with distinct features by the enhancement model through generation. The subsequent segmentation task is thereby simplified, leading to higher accuracy. We design a lightweight generator and discriminator to improve both enhancement quality and computational efficiency. Extensive experiments demonstrate the effectiveness of each component within GeNSeg-Net. Compared to existing methods, GeNSeg-Net achieves state-of-the-art (SOTA) segmentation accuracy and generalization performance on both private and public datasets, while maintaining highly competitive processing speed. **Code will be available for research and clinical purposes.**

## CCS CONCEPTS

• **Computing methodologies** → **Image representations**; • **Applied computing** → **Health informatics**.

## KEYWORDS

immunohistochemistry images; nucleus segmentation; generative adversarial network

## 1 INTRODUCTION

Immunohistochemistry (IHC) staining employs the specificity of antigen-antibody reactions to visualize specific protein expression within tissues or cells. Its high specificity and sensitivity, along with the ability for multi-target quantification and localization, have contributed to its growing popularity [2, 12, 35, 37]. It encompasses

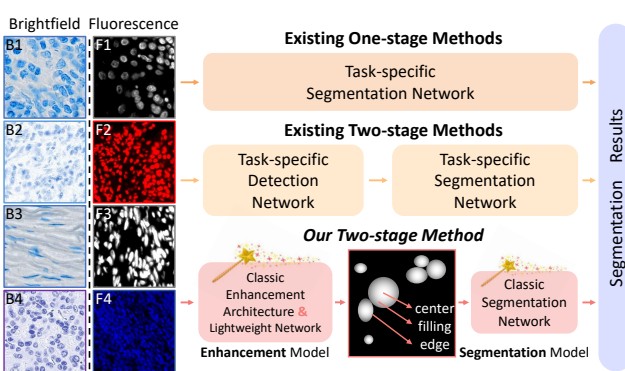

**Figure 1: The diagram of our method. The left two columns display IHC images, each sourced from a different tissue type. B1, B2, B3 and B4 are brightfield images, while those prefixed with "F" are fluorescence images. The image generated by enhancement model exemplifies the "ideal" nuclei characterized by clear centers, solid filling and well-defined edges. Best viewed with zoom-in.**

two major imaging techniques: brightfield and fluorescence. Accurate nucleus segmentation in IHC images, involving detection and delineation of each nucleus, is vital for disease diagnosis and treatment [4]. For instance, the classification and grading of cancers heavily rely on the rich spatial and morphological information of nuclei [25]. Manual nucleus segmentation is time-consuming, which means the increasing need for automated segmentation.

Automated nucleus segmentation remains challenging due to the inherent characteristics of nuclei, i.e., high density and adhesive edges, as depicted in Fig. 1, resulting in under- or over-segmentation. Furthermore, complex factors such as variable shapes and sizes, blurred edges, overlapping cell clusters, uneven staining and imaging condition variations lead to high error rates [11]. A large number of traditional and deep learning-based methods have been proposed to address above problems [5, 9, 15, 23, 27, 34]. In IHC images, there are different imaging techniques, i.e., brightfield and fluorescence, and significant morphological variations originating from diverse tissues. Existing methods often struggle with generalization ability. They exhibit poor performance on nucleus segmentation for tissue types not included in the training data. Since encompassing all morphological variations during training is impractical, our aim is to conduct a pilot study to develop a general framework for nucleus segmentation in IHC images. By training on several tissue types captured via brightfield and fluorescence imaging, the method achieves accurate nucleus segmentation for various types, regardless of their presence during training.

Fig. 1 showcases significant variations in nucleus morphology across diverse tissue types in both IHC brightfield and fluorescence images. For instance, in brightfield images, nuclei in image B1 exhibit a solid texture while those in B4 display a hollow texture. Nuclei in B4 are relatively regular ellipses, whereas those in B1,

B2 and B3 show irregular shapes. Hence, we ponder: Is it possible to minimize these morphological variations arising from various factors mentioned before? If so, the subsequent segmentation task is simplified when dealing with nuclei of uniform morphology. This leads to higher segmentation accuracy across various tissue types, thereby improving the method's generalization performance.

In this paper, we propose a novel two-stage method as illustrated in Fig. 1. It deviates from the common approach of existing two-stage methods [1, 10, 16], which involves initial nucleus detection followed by precise segmentation. Inspired by the advancements in Artificial Intelligence Generated Content (AIGC) [6, 45], nuclei of various tissue types and imaging techniques are enhanced to an "ideal" morphology characterized by clear centers, solid filling and well-defined edges through generation in enhancement model. Subsequently, the nuclei with distinct features in texture and edges are input into the segmentation model to obtain precise contours. In contrast to existing one-stage [9, 15, 33] and two-stage [1, 10, 16] deep learning methods with task-specific network designs, a classic image-to-image translation generative adversarial network (GAN) [14] performs enhancement alongside a conventional network for segmentation. Since two-stage methods which are not designed to be end-to-end increase complexity to the entire framework [9], we develop a lightweight generator and discriminator based on ResNet and transformer [38], respectively, to improve both enhancement quality and computational efficiency.

The main contributions of this paper are as follows:

- This paper is the pioneering study on the general framework for nucleus segmentation in IHC images, with a focus on high generalization ability. We aim to accurately segment nuclei across various tissue types in both brightfield and fluorescence images by training on a small subset of types.
- We propose a novel two-stage method, GeNSeg-Net, which first enhances nuclei followed by segmentation. This process generates nuclei of various tissues with clearer texture and edges, effectively mitigating segmentation challenges.
- We design a lightweight generator and discriminator to improve enhancement quality by emphasizing the semantic relationship, texture, shape and size of generated nuclei further, while ensuring computational efficiency.
- We conduct experiments on a private systematic dataset of nuclei, covering diverse tissues with rich stain colors in both brightfield and fluorescence imaging, as well as public datasets DSB2018 [4] and BBBC006v1 [24] which include fluorescence images. Our method, GeNSeg-Net, exhibits SOTA performance in accuracy.

## 2 RELATED WORK

In this section, we divide current nucleus segmentation methods into traditional methods and deep learning-based methods.

Many traditional methods are based on watershed [27, 36, 39, 46]. For instance, Malpica et al. [27] introduced a morphological watershed-based method that utilizes both intensity and morphological information for nucleus segmentation. However, it tends to result in under- or over-segmentation when dealing with nucleus adhesion [36, 46]. Yang et al. [46] proposed a novel marker extraction method based on condition erosion to mitigate over-segmentation. Additionally, numerous other methods exist, such as threshold-based [5], contour-based [28], graph-based [3], region-based [40]

and others [26, 34, 43]. A common drawback of traditional methods is their reliance on manual feature extraction, which leads to good performance solely on specific datasets with rich features.

In recent years, deep learning-based methods have gained prominence. They can be categorized into one-stage [7, 9, 15, 22, 33] and two-stage [1, 10, 16, 23] methods. One-stage methods employ a single network and use post-processing to obtain precise contours, which can be further divided into classification-based and regression-based methods. Classification-based methods output classification probability maps. For example, DCAN [7] is a deep contour-aware network that predicts nuclei and boundaries simultaneously through two branches: a semantic segmentation branch and a boundary detection branch. BES-Net [29] and CIA-Net [49] establish connections between the two branches, further improving effectiveness. HARU-Net [8], the latest advancement, utilizes a hybrid attention-based residual U-blocks network to predict foreground regions and boundaries simultaneously. These methods derive final nucleus instances by subtracting boundaries from foreground regions, potentially resulting in certain loss of segmentation accuracy. Regression-based methods output regression maps. For instance, HoVer-Net [15] predicts the distances between nucleus pixels and their centroids in both the vertical and horizontal directions, followed by watershed post-processing. StarDist [33] predicts centroid probability maps and distances from each foreground pixel to its instance boundary along pre-defined directions. However, StarDist may loss information for large nuclei due to its reliance on only central pixel features in post-processing, which is addressed by CPP-Net [9]'s optimization. Two-stage methods typically involve a detection stage followed by a segmentation stage, i.e., first locating nucleus instances and then predicting precise masks. For example, BRP-Net [10] generates nucleus proposals based on instance boundaries and then refines the foreground masks. SAM [20] represents the emergence of large-scale segmentation models, excelling in natural image segmentation tasks, but its performance on special objects like nucleus is yet to be verified.

## 3 METHODOLOGY

The overall architecture of GeNSeg-Net is depicted in Fig. 2. It consists of two stages: an enhancement stage and a segmentation stage. In the first stage, the texture and edges of all nuclei are enhanced to obtain a uniform morphology with distinct features. The lightweight generator and discriminator in enhancement model improves both the quality of enhancement and computational efficiency. In the second stage, we perform class prediction and post-processing for segmentation.

### 3.1 Data Pre-processing

Data pre-processing is illustrated in the red area of Fig. 2. Considering texture and edge variations among nuclei, we initially enhance them through GAN [14, 17], achieving consistent "ideal" nuclei with clear centers, solid filling and well-defined edges, referred to as "nucleus enhancement". To obtain the ground-truth ideal nuclei, we calculate the pixel-to-boundary Euclidean distance for each nucleus in the annotated images. This categorizes each pixel into three classes: nucleus bodies, pending edges and background. More specifically, we adjust the staining intensity of each pixel within a

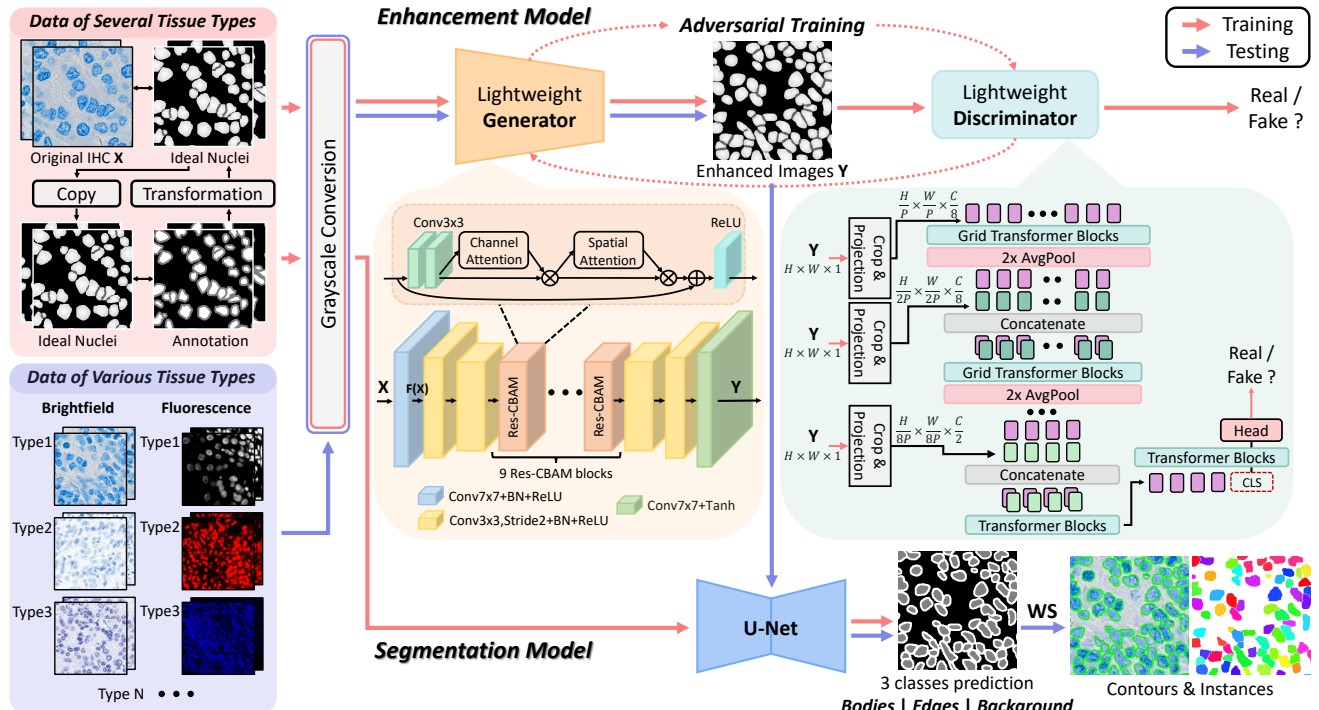

**Figure 2: The overview of our proposed GeNSeg-Net. We take high-resolution images sized at** $512 \times 512$ **pixels as a typical example to illustrate our method. "WS" denotes watershed segmentation. For clarity, we display images with a size of** $256 \times 256$ **pixels and bold the precise contours generated by watershed.**

nucleus based on its distance from the centroid, gradually decreasing towards the edge. Shared boundaries among nuclei receive a same staining intensity, termed pending edges. The transformed data serves as ground truth for the first stage and as input for the second stage.

## 3.2 Enhancement Model

*3.2.1 Lightweight Generator.* For the relatively straightforward generation task, we design a lightweight network as depicted in Fig. 2. Initially, the high-resolution image $\mathbf{X}$ is transformed into a feature map $\mathbf{F}(\mathbf{X})$ with $C$ channels using the convolution operation. Subsequently, spatial information of each nucleus is aggregated at multiple levels through downsampling. The aggregation effectively consolidates nucleus pixels while eliminating isolated background noise. Simultaneously, applying subsequent operations to the downsampled feature map improves computational efficiency. The feature map then passes through 9 custom-designed Res-CBAM blocks. In Res-CBAM blocks, channel attention and spatial attention are integrated after convolution to eliminate irrelevant information and noise [8, 44]. Finally, the feature map is gradually upsampled to the original size, yielding the enhanced image $\mathbf{Y}$. The process mitigates the loss of fine details caused by direct upsampling at large ratios. In the experiment, we meticulously assess the impact of downsampling and Res-CBAM on model's performance.

*3.2.2 Lightweight Discriminator.* In modern GANs [31], the discriminator typically employs convolutional neural networks (CNN) as backbone. While it facilitates stable training for high-resolution images, the convolutional operation has a limited local receptive

field. In cases with extensive nucleus adhesion or large nuclei, insufficient network depth fails to capture long-range dependencies. However, increasing network depth can lead to feature and detail loss, complicating optimization and reducing computational efficiency, as demonstrated in previous studies [41, 47]. Therefore, we deviate from conventional CNN models while conducting a comparative analysis in our experiments.

In addition, we consider the following factors: (1) A nucleus covers multiple pixels, especially in cases of adhesive nuclei that span a significant area. Hence, rather than pixel-level discrimination, we opt for a coarser patch level. (2) Large patches in high-resolution images tend to lose low-level texture details, while small patches significantly increase computational demands and memory usage. (3) The adhesion and size of nuclei vary, making it challenging to adopt a fixed patch size for the entire image, which can affect model generalization.

Inspired by TransGAN [18], we introduce a transformer-based multi-scale lightweight discriminator, employing the transformer encoder [38] as the basic block. Our multi-scale discriminator handles local semantic relationships among nuclei, whether they are isolated or adhesive, regardless of their size. It also considers low-level texture, as well as global shape and size features by employing patches of varying sizes at different scales. In high-resolution feature maps, correlating two distant positions is unnecessary. Thus, we integrate grid self-attention [18] into the transformer block to improve computational efficiency.

In Fig. 2, the generator produces an enhanced image $\mathbf{Y}$, which is then fed into the discriminator. At various stages, $\mathbf{Y}$ of size $H \times W \times 1$ is cropped into four sequences by selecting different patch size $P$,

$2P$, $4P$ and $8P$. Considering the typical size of nuclei, the patch size can be set to 8. Subsequently, these sequences are linearly projected into dimensions $(\frac{H}{P} \times \frac{W}{P}) \times \frac{C}{8}$, $(\frac{H}{2P} \times \frac{W}{2P}) \times \frac{C}{8}$, $(\frac{H}{4P} \times \frac{W}{4P}) \times \frac{C}{4}$ and $(\frac{H}{8P} \times \frac{W}{8P}) \times \frac{C}{2}$. These sequences, combined with learnable positional encoding, serve as inputs to the transformer blocks. After each block, features are upsampled via average pooling and concatenated with the new sequence. A class token is introduced prior to the final transformer. It passes through the transformer block to determine whether Y is real or fake.

Grid self-attention [18] is adopted in high-resolution stages (resolution higher than $16 \times 16$). In contrast to conventional transformer blocks [13] that involve interactions between the individual token and all others, we partition the entire feature map into grids based on a pre-defined window size. Attention operation is then performed within each grid, tailored to our requirements.

### 3.3 Segmentation Model

In the second stage, class prediction is conducted firstly by U-Net [32], followed by watershed post-processing. To be specific, U-Net categorizes the image context into three classes: nucleus bodies, coarse edges (approximately 4 pixels wide) and background. The watershed algorithm then defines contours within the coarse edge regions. Our segmentation model follows a classic approach with a simple network design. It achieves both high accuracy and computational efficiency. Since the segmentation model operates independently of the enhancement model, GeNSeg-Net provides great flexibility, enabling the integration of more advanced segmentation methods. Despite such a classic method within our framework, the segmentation accuracy has already achieved SOTA performance.

### 3.4 Training and Inference

Our training data consists of several tissue types, while the testing data includes nuclei from both the same tissue types as in the training and different tissue types. To focus the networks' attention on nucleus morphology and minimize interference from imaging techniques and other factors, both the training and testing data are uniformly converted into grayscale images, as shown in the gray area of Fig. 2. The enhancement model and segmentation model are trained individually using their respective paired data for full supervision.

During the training of our enhancement model, the loss function for the image-to-image translation GAN is defined as:

$$L_{GAN}(G, D) = \mathbb{E}_{y \sim p_{data}(y)}\{log D(y)\} \\ + \mathbb{E}_{x \sim p_{data}(x)}\{log(1 - D(G(x)))\} \quad (1)$$

where $G$ and $D$ refer to the generator and discriminator, while $x$ and $y$ represent the input and ground truth, respectively. Previous research suggests that introducing noise to the input can prevent deterministic outputs and address issues related to fitting restricted distributions [42]. However, nucleus images inherently contain background noise due to various factors. The intensity and distribution of pixel values are crucial for the model to learn localization and texture, with subtle variations playing a significant role during learning. Therefore, we refrain from introducing additional information besides the original input. Nevertheless, incorporating a traditional loss, such as L1, has proven beneficial for model

training [30]. While the discriminator's task remains unchanged, the generator now has a dual objective: to generate images with realism in the semantic relationship, texture, shape and size, and to approach the ground truth in an L1 sense. The overall loss function of enhancement model is:

$$L_T(G) = \mathbb{E}_{x,y}\{\| y - G(x) \|_1\} \quad (2)$$

$$L_{enh}(G, D) = L_T(G) + \lambda L_{GAN}(G, D) \quad (3)$$

In the fully supervised U-Net training, a combined Cross Entropy and Dice loss function is utilized for predicting three classes. The segmentation model's loss function is formulated as:

$$L_{seg} = \alpha \cdot L_{CE} + \beta \cdot L_{Dice} \quad (4)$$

The weight of each term is controlled by $\alpha$ and $\beta$.

During inference, test images of diverse tissue types initially undergo enhancement by the trained generator. Subsequently, the trained U-Net model predicts the three classes. Precise contours are then obtained by watershed post-processing.

## 4 EXPERIMENTAL SETUP

### 4.1 Dataset

*DSB2018 and BBBC006v1.* Data Science Bowl 2018 (DSB2018) [4] is a competition focused on nucleus detection and segmentation. It comprises 670 images with manual annotations, sized from 256×256 to 520×696 pixels. Fluorescence imaging is utilized, encompassing DAPI and Hoechst 33342 stains. We follow the dataset division protocol outlined in [9], allocating 380 images for training, 67 for validation and 50 for testing. BBBC006v1 [24] consists of 768 fluorescence images containing Hoechst 33342-stained U2OS cells, each with the size 696×520 pixels. The dataset is randomly divided into 462 training images, 153 validation images and 153 testing images.

*Private Systematic Dataset.* To address the lack of a systematic public IHC dataset, we build a nucleus dataset encompassing diverse tissue types and quantities. It consists of IHC images acquired through two primary imaging techniques: brightfield and fluorescence. In brightfield imaging, hematoxylin stains nuclei, yielding vibrant images under transmitted light. Fluorescence imaging employs DAPI to label nuclei. The nuclei emit fluorescence signal when excited at specific wavelengths, which is then captured using a fluorescence microscope. To emulate real-world scenarios, we refrain from strictly controlling staining and imaging conditions, allowing for natural variations introduced by experts and environments. In clinical and industrial settings, pseudo-colors are commonly applied to fluorescence images for better visualization. Hence, we randomly assign pseudo-colors (e.g. red, blue and orange) while ensuring consistent color assignment within the same tissue. All stained whole slide images are segmented into patches of 512×512 pixels. Five pathologists with over three years of clinical experience provided pixel-wise annotations, as illustrated in Fig. 3. In the training dataset, images are annotated into three classes: nucleus bodies, edges and background. To mitigate annotation discrepancies among experts, edges are required to be coarsely annotated with a 4-pixel width, covering the entire actual edges. Two pixels are within the nuclei, while the other two pixels lie outside. In the testing dataset, each instance is precisely annotated.

The dataset composition and division are summarized in supplementary materials. Training and validation data originates from four tissue types: lung, liver, colon and stomach cancers, each with 60 images, alongside 20 testing images per tissue. Additionally, the testing data includes four different tissues: cervix, osteosarcoma, tuberculoma and lymph cancers, each comprising 20 images. They serve to evaluate the generalization ability of segmentation methods. During training, both brightfield and fluorescence data are combined, resulting in a total of 480 images. Among these, 384 images are randomly selected for training, with the remaining 96 reserved for validation.

It's noteworthy that our private dataset, compared to DSB2018 [9] and BBBC006v1 [24], contains a larger number of nuclei with more diverse morphology and more complex environment, as depicted in Fig. 4 and 5. As a result, the segmentation task is significantly more challenging.

## 4.2 Evaluation Metrics

We employ six evaluation metrics to fully assess method performance: (1) Dice: It evaluates the separation of nuclei from the background. (2) Aggregated Jaccard Index (AJI) [21] and AJI+: AJI is based on instance-wise IoU between predictions and ground truth [10, 15, 48]. AJI+ improves AJI by ensuring maximal unique pairing to obtain overall intersection, thus addressing AJI's over-penalisation. (3) Panoptic Quality (PQ) [19]: It has been widely employed in panoptic segmentation tasks [19]. PQ was first introduced into nucleus segmentation by [15] as the most comprehensive and persuasive metric. It evaluates both detection quality (DQ) and instance-wise segmentation quality (SQ) comprehensively.

## 4.3 Implementation Details

All methods are implemented using PyTorch 1.12.1 on a system equipped with a single NVIDIA GeForce RTX 3090 GPU. During training, we maintain the original image size and apply basic data augmentation techniques for all methods, including flip, rotation and brightness adjustment. For GeNSeg-Net, the enhancement model adopts Adam optimization with a learning rate of $10^{-4}$. A linear decay strategy is employed in the latter half of epochs. Batch size is set to 2 and training proceeds for 400 epochs. Considering both nucleus size and computational efficiency, the discriminator's window size is configured to 8. The loss weight coefficient $\lambda$ is empirically set to 0.01. The enhancement model's backbone is Pix2pix [17] without any condition. In the segmentation model, the U-Net architecture is standard as proposed in [32], with a fixed learning rate of $10^{-2}$. $\alpha$ and $\beta$ are respectively set to 1 and 3. For other methods, we adhere to the settings in their respective papers.

## 5 EXPERIMENTS AND ANALYSIS

## 5.1 Comparisons with the SOTA Methods

To evaluate both the segmentation accuracy and speed of our proposed GeNSeg-Net, we conduct experiments on our private systematic dataset, as well as public datasets DSB2018 [4] and BBBC006v1 [24]. GeNSeg-Net is compared against a large-scale segmentation model and SOTA nucleus segmentation methods, i.e., SAM [20], NucleiSegNet [22], HoVer-Net [15], StarDist [33] and CPP-Net [9]. To ensure fair comparison, we follow the same data split protocol

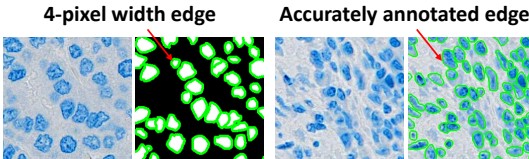

**4-pixel width edge**    **Accurately annotated edge**

**Figure 3: Annotation examples of brightfield training data (left) and testing data (right).**

as outlined in Section 4.1 and apply consistent data augmentation techniques for all methods. During training, for methods other than GeNSeg-Net, we adopt the midpoint position of the 4-pixel width annotation as ground truth since they require precise edge annotation.

*5.1.1 Comparisons on the Private Systematic Dataset.* In Table 1, the results of brightfield same tissue testing are shown. Notably, NucleiSegNet [22], HoVer-Net [15], StarDist [33] and CPP-Net [9] exhibit robust performance. SAM [20] exhibits advantages in natural images but tends to under-segment due to the significant adhesion of nuclei and minimal target variations in our task. Moreover, SAM fails to identify nuclei of low signal and blurred edges, leading to increased false negatives. NucleiSegNet's well-designed architecture facilitates efficient object localization and high-level semantic map extraction while consciously mitigating false positives and negatives. HoVer-Net performs well in identifying foreground regions. It effectively addresses adhesive and blurred nuclei through distance-based predictions. Although StarDist is proficient in foreground region identification and adhesive nuclei differentiation, its instance metrics are compromised due to the boundary influence. StarDist's reliance solely on central pixel features results in incomplete contours, particularly obvious for large nuclei. Additionally, its inclination towards representing nucleus shapes with relatively regular ellipses leads to inaccuracies when depicting irregularly shaped nuclei. CPP-Net improves upon StarDist by more accurately identifying foreground nuclei, distinguishing adhesive nuclei and representing diverse shapes. GeNSeg-Net outperforms these methods across all metrics, with improvements of approximately 2.6% in AJI, 3.1% in AJI+, 0.1% in Dice, 3.1% in DQ, 2.7% in SQ and 5.7% in PQ. The quantitative comparison underscores its superior accuracy.

In brightfield different tissue testing, notable variations in nucleus morphology not presented during training pose segmentation challenges. Testing metrics are presented in Table 1. NucleiSegNet [22] and HoVer-Net [15] exhibit poor segmentation performance. In contrast, StarDist [33] and CPP-Net [9] show superior generalization ability, albeit with a performance decrease ranging from 4% to 13% across metrics. The decline is mainly due to their difficulty in detecting nuclei with different texture, shapes, sizes and signal strength. CPP-Net's reliance on similar nucleus shapes as a design prior further constrains its performance on datasets with diverse shapes. GeNSeg-Net maintains robustness and demonstrates superior results across all metrics compared to current SOTA methods.

In fluorescence tissue testing, most methods surpass their brightfield counterparts as shown in Table 2. This superiority arises from differences in staining and imaging, which result in fluorescence images with reduced background noise, heightened signal intensity and clearer nucleus edges compared to brightfield images. As a result, while maintaining consistent trends with brightfield testing,

**Table 1: Comparisons on private brightfield data.**

| Methods | Metrics | | | | | |
|---|---|---|---|---|---|---|
| | AJI↑ | AJI+↑ | Dice↑ | DQ↑ | SQ↑ | PQ↑ |
| Brightfield same tissue testing | | | | | | |
| SAM [20] | 0.561 | 0.592 | 0.854 | 0.748 | 0.795 | 0.595 |
| NucleiSegNet [22] | 0.700 | 0.730 | 0.877 | 0.886 | 0.823 | 0.729 |
| HoVer-Net [15] | 0.654 | 0.690 | 0.840 | 0.852 | 0.830 | 0.707 |
| StarDist [33] | 0.655 | 0.684 | 0.850 | 0.857 | 0.800 | 0.686 |
| CPP-Net [9] | 0.700 | 0.722 | 0.873 | 0.875 | 0.826 | 0.723 |
| GeNSeg-Net (ours) | **0.726** | **0.761** | **0.878** | **0.917** | **0.857** | **0.786** |
| Brightfield different tissue testing | | | | | | |
| SAM [20] | 0.470 | 0.494 | 0.713 | 0.590 | 0.736 | 0.434 |
| NucleiSegNet [22] | 0.423 | 0.454 | 0.749 | 0.479 | 0.746 | 0.357 |
| HoVer-Net [15] | 0.422 | 0.457 | 0.575 | 0.589 | 0.753 | 0.444 |
| StarDist [33] | 0.591 | 0.624 | 0.758 | 0.811 | 0.737 | 0.598 |
| CPP-Net [9] | 0.587 | 0.620 | 0.749 | 0.801 | 0.750 | 0.601 |
| GeNSeg-Net (ours) | **0.701** | **0.729** | **0.783** | **0.895** | **0.843** | **0.754** |

**Table 2: Comparisons on private fluorescence data.**

| Methods | Metrics | | | | | |
|---|---|---|---|---|---|---|
| | AJI↑ | AJI+↑ | Dice↑ | DQ↑ | SQ↑ | PQ↑ |
| Fluorescence same tissue testing | | | | | | |
| SAM [20] | 0.363 | 0.423 | 0.772 | 0.442 | 0.843 | 0.373 |
| NucleiSegNet [22] | 0.500 | 0.541 | **0.892** | 0.632 | 0.808 | 0.511 |
| HoVer-Net [15] | 0.615 | 0.682 | 0.842 | 0.833 | 0.853 | 0.711 |
| StarDist [33] | 0.677 | 0.722 | 0.869 | 0.902 | 0.819 | 0.739 |
| CPP-Net [9] | 0.709 | 0.753 | 0.875 | **0.944** | 0.833 | 0.786 |
| GeNSeg-Net (ours) | **0.737** | **0.779** | 0.886 | 0.936 | **0.863** | **0.808** |
| Fluorescence different tissue testing | | | | | | |
| SAM [20] | 0.328 | 0.448 | 0.826 | 0.553 | 0.800 | 0.442 |
| NucleiSegNet [22] | 0.526 | 0.560 | 0.882 | 0.628 | 0.764 | 0.48 |
| HoVer-Net [15] | 0.636 | 0.690 | 0.832 | 0.817 | 0.839 | 0.685 |
| StarDist [33] | 0.674 | 0.713 | 0.863 | 0.869 | 0.799 | 0.694 |
| CPP-Net [9] | 0.695 | 0.736 | 0.875 | 0.889 | 0.816 | 0.725 |
| GeNSeg-Net (ours) | **0.721** | **0.766** | **0.891** | **0.900** | **0.848** | **0.763** |

metrics in fluorescence different tissue testing exhibit no significant decrease compared to those in fluorescence same tissue testing. Overall, GeNSeg-Net maintains an advantage over other methods.

*5.1.2 Comparisons on the DSB2018 and BBBC006v1 Dataset.* In contrast to our private dataset, images in DSB2018 [4] and BBBC006v1 [24] feature nuclei with lower density and more consistent morphology. This leads to the decent performance of HoVer-Net [15], StarDist [33] and CPP-Net [9], as presented in Table 3. Compared to the testing results of private fluorescence data, nearly all metrics show improvement. However, upon closer examination, we notice the presence of adhesive nuclei that these methods fail to fully distinguish. Our method effectively reduces false positives and negatives, achieving higher segmentation accuracy.

**Table 3: Comparisons on DSB2018 and BBBC006v1.**

| Methods | Metrics | | | | | |
|---|---|---|---|---|---|---|
| | AJI↑ | AJI+↑ | Dice↑ | DQ↑ | SQ↑ | PQ↑ |
| DSB2018 | | | | | | |
| SAM [20] | 0.674 | 0.698 | **0.941** | 0.798 | 0.859 | 0.693 |
| NucleiSegNet [22] | 0.662 | 0.689 | 0.934 | 0.787 | 0.848 | 0.678 |
| HoVer-Net [15] | 0.777 | 0.788 | 0.905 | 0.868 | 0.877 | 0.767 |
| StarDist [33] | 0.795 | 0.806 | 0.915 | 0.902 | 0.850 | 0.770 |
| CPP-Net [9] | 0.832 | 0.843 | 0.936 | 0.934 | 0.873 | 0.818 |
| GeNSeg-Net (ours) | **0.841** | **0.862** | 0.930 | **0.940** | **0.882** | **0.829** |
| BBBC006v1 | | | | | | |
| SAM [20] | 0.701 | 0.727 | 0.935 | 0.845 | 0.861 | 0.727 |
| NucleiSegNet [22] | 0.638 | 0.688 | 0.943 | 0.793 | 0.868 | 0.690 |
| HoVer-Net [15] | 0.915 | 0.921 | 0.976 | 0.961 | 0.958 | 0.920 |
| StarDist [33] | 0.914 | 0.918 | 0.970 | 0.966 | 0.950 | 0.917 |
| CPP-Net [9] | 0.961 | 0.963 | **0.993** | **0.990** | 0.978 | **0.969** |
| GeNSeg-Net (ours) | **0.968** | **0.969** | 0.991 | 0.988 | **0.981** | 0.969 |

**Table 4: Cross-dataset evaluation.**

| Tasks | Methods | Metrics | | | | | |
|---|---|---|---|---|---|---|---|
| | | AJI↑ | AJI+↑ | Dice↑ | DQ↑ | SQ↑ | PQ↑ |
| Private ↓ DSB2018 | HoVer-Net[15] | 0.727 | 0.735 | 0.843 | 0.848 | 0.826 | 0.711 |
| | StarDist[33] | 0.764 | 0.773 | 0.895 | 0.917 | 0.814 | 0.747 |
| | CPP-Net[9] | 0.823 | 0.831 | 0.925 | **0.943** | 0.850 | 0.802 |
| | GeNSeg-Net | **0.837** | **0.845** | **0.935** | 0.929 | **0.868** | **0.808** |
| Private ↓ BBBC 006v1 | HoVer-Net[15] | 0.775 | 0.778 | 0.92 | 0.901 | 0.850 | 0.766 |
| | StarDist[33] | 0.763 | 0.765 | 0.914 | 0.879 | 0.838 | 0.752 |
| | CPP-Net[9] | 0.790 | 0.792 | 0.929 | **0.910** | 0.855 | 0.778 |
| | GeNSeg-Net | **0.801** | **0.803** | **0.933** | 0.899 | **0.868** | **0.781** |
| DSB2018 ↓ BBBC 006v1 | HoVer-Net[15] | 0.754 | 0.756 | 0.908 | 0.868 | 0.837 | 0.727 |
| | StarDist[33] | 0.762 | 0.763 | 0.908 | 0.876 | 0.839 | 0.735 |
| | CPP-Net[9] | 0.765 | 0.766 | 0.915 | 0.883 | 0.846 | 0.748 |
| | GeNSeg-Net | **0.784** | **0.789** | **0.923** | **0.889** | **0.863** | **0.767** |
| BBBC 006v1 ↓ DSB2018 | HoVer-Net[15] | 0.570 | 0.588 | 0.812 | 0.660 | 0.784 | 0.531 |
| | StarDist[33] | 0.537 | 0.558 | 0.792 | 0.644 | 0.762 | 0.504 |
| | CPP-Net[9] | 0.426 | 0.434 | 0.666 | 0.469 | 0.729 | 0.378 |
| | GeNSeg-Net | **0.665** | **0.673** | **0.821** | **0.755** | **0.836** | **0.631** |

*5.1.3 Cross-dataset Evaluation.* To further assess GeNSeg-Net's generalization performance, we design cross-dataset evaluation experiments. Specifically, we evaluate GeNSeg-Net with HoVer-Net [15], StarDist [33] and CPP-Net [9] across four tasks: our private dataset (Private)→DSB2018 [4], Private→BBBC006v1 [24], DSB2018→BBBC006v1 and BBBC006v1→DSB2018. Here, datasets listed before the arrow represent the training data, while those listed after denote the testing data. We present results in Table 4. The metrics reveal that models trained on our private dataset and DSB2018, which encompass diverse tissue types, exhibit superior generalization ability. Conversely, the limited diversity of tissue types in BBBC006v1 impedes the model's generalization ability. Across all four tasks, GeNSeg-Net consistently outperforms other

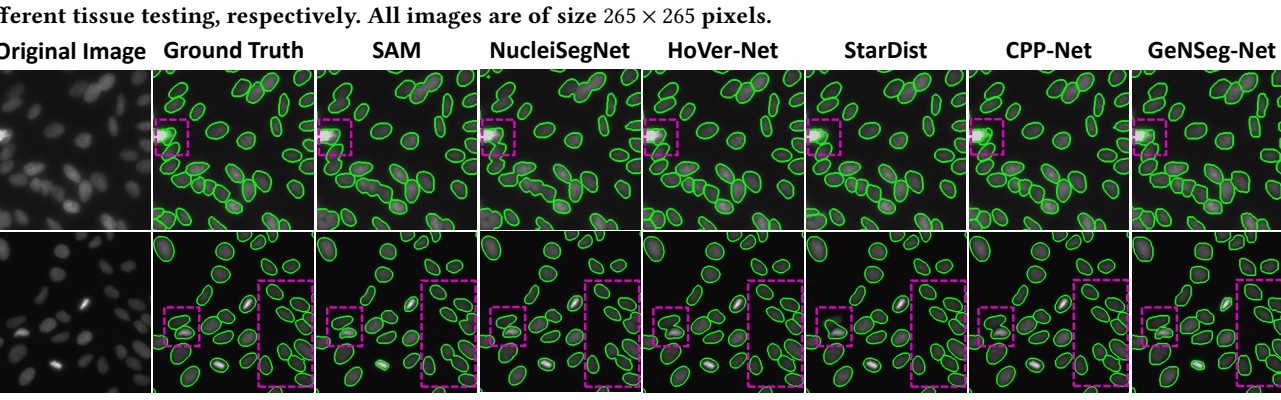

**Figure 4: Qualitative comparisons with SOTA methods on our private dataset. The four lines from top to bottom present images from brightfield same tissue testing, brightfield different tissue testing, fluorescence same tissue testing and fluorescence different tissue testing, respectively. All images are of size $265 \times 265$ pixels.**

**Figure 5: Qualitative comparisons with SOTA methods on the DSB2018 [4] and BBBC006v1 [24] dataset. The two lines from top to bottom present images from DSB2018 and BBBC006v1, respectively. All images are of size $265 \times 265$ pixels.**

methods. For instance, in the DSB2018→BBBC006v1 task, GeNSeg-Net exhibits improvements of 1.9%, 2.3%, 0.8% and 1.9% in AJI, AJI+, Dice and PQ, respectively. It's noteworthy that GeNSeg-Net significantly improves all metrics in the BBBC006v1→DSB2018 task, though the scores remain below those in the Private→DSB2018 task. This highlights the importance of increasing the diversity of tissue types in the training data, even if the types in testing are not present during training.

*5.1.4 Inference Time.* We evaluate the inference speed of all methods on our private dataset using the same computer as outlined in Section 4.1 and 4.3. Applying same post-processing method to $512 \times 512$ images, the average inference time per image for SAM [20], NucleiSegNet [22], HoVer-Net [15], StarDist [33], CPP-Net [9] and GeNSeg-Net is 1.938, 0.548, 1.863, 0.203, 0.353 and 0.242 seconds, respectively, covering both prediction and post-processing

time. Obviously, StarDist exhibits the fastest inference speed. Despite being a two-stage method, GeNSeg-Net's lightweight design enables faster processing compared to most existing methods.

## 5.2 Qualitative Comparisons

In this section, we qualitatively compare all methods on our private dataset, DSB2018 [4] and BBBC006v1 [24]. Results for the private dataset are shown in Fig. 4, while those for DSB2018 and BBBC006v1 are in Fig. 5, in line with our previous analysis. GeNSeg-Net, incorporating the enhancement model, excels in identifying weakly stained nuclei, as indicated by the red boxes in Fig. 4. It also effectively distinguishes adhesive nuclei and reduces over-segmentation, as shown by the purple boxes in Fig. 4 and 5. Furthermore, thanks to our segmentation model, GeNSeg-Net produces more natural contours, better capturing the true shapes of nuclei.

**Table 5: Ablation study of each component in GeNSeg-Net.**

| Enh model | | Seg model | Metrics | | | | | |
|---|---|---|---|---|---|---|---|---|
| Gen | Dis | | AJI↑ | AJI+↑ | Dice↑ | DQ↑ | SQ↑ | PQ↑ |
| – | – | ✓ | 0.580 | 0.650 | 0.820 | 0.762 | 0.844 | 0.643 |
| CPP-Net [9] | | | 0.673 | 0.708 | 0.843 | 0.877 | 0.806 | 0.707 |
| basic | basic | ✓ | 0.682 | 0.715 | 0.844 | 0.853 | 0.835 | 0.712 |
| basic | ours | ✓ | 0.685 | 0.732 | 0.855 | 0.872 | 0.852 | 0.743 |
| ours | ours | ✓ | **0.721** | **0.759** | **0.860** | **0.912** | **0.853** | **0.778** |

**Table 6: Ablation study on the generator.**

| Gen | | Dis | Metrics | | | | | |
|---|---|---|---|---|---|---|---|---|
| Backbone | DS US | | AJI↑ | AJI+↑ | Dice↑ | DQ↑ | SQ↑ | PQ↑ |
| basic | – | ours | 0.685 | 0.732 | 0.855 | 0.872 | 0.852 | 0.743 |
| ResNet9 | ✓ | ours | 0.711 | 0.755 | 0.852 | 0.908 | **0.853** | 0.775 |
| Res-CBAM | ✓ | ours | **0.721** | **0.759** | **0.860** | **0.912** | **0.853** | **0.778** |

## 5.3 Ablation Study

We conduct ablation studies on all brightfield and fluorescence testing data in our private dataset. Initially, the efficacy of the enhancement model is demonstrated. We then assess the discriminator's effectiveness and determine the generator's structure. Qualitative comparisons depicted in Fig. 6 show enhanced images in the first row, predicted class maps in the second row and segmentation results in the final row, with the top-left image representing the ground truth. Evaluation metrics from the ablation study of each component in GeNSeg-Net and the generator are presented in Table 5 and 6, where "gen", "dis", "enh model", "seg model", "ds" and "us" represent the generator, discriminator, enhancement model, segmentation model, downsampling and upsampling, respectively.

*5.3.1 Ablation Study of Each Component in GeNSeg-Net.* When solely relying on the segmentation model, it effectively identifies foreground nucleus regions, as indicated by the relatively high Dice. However, due to the network's general design and lack of prior information, it struggles to distinguish adhesive nuclei, resulting in noticeable under-segmentation, as shown in the first column of Fig. 6. To address this issue, we introduce an enhancement model aimed at enhancing morphological features, i.e., ensuring clear centers, solid filling and well-defined edges to simplify subsequent segmentation. Initially, the enhancement model employs ResNet as the generator's backbone, comprising 9 blocks, denoted as "basic" in Table 5 and 6. We adopt a CNN-based discriminator proposed in [17], which discriminates at the patch level, as the "basic" discriminator. Integrating the enhancement model significantly improves segmentation performance, with AJI, AJI+, Dice, DQ, SQ and PQ reaching 0.682, 0.715, 0.844, 0.853, 0.835 and 0.712, respectively, slightly surpassing CPP-Net [9]. In our straightforward generation task, increasing the number of blocks adds complexity to training, while reducing blocks leads to insufficient model learning, reflected in declining evaluation metrics. We further attempt to employ a standard U-Net [32] as the generator's backbone, but it results in enhanced images with increased artifacts and noise. It performs worse in distinguishing the two foreground classes, i.e., nucleus

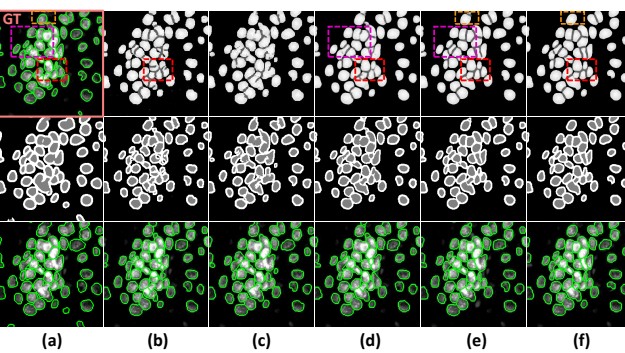

**Figure 6: Qualitative comparisons in the ablation study. The six columns from left to right represent -/- (a), basic/basic (b), U-Net/basic (c), basic/ours (d), ResNet9 DS&US/ours (e) and Res-CBAM DS&US/ours (f). Left and right of "/" indicate the structure of generator and discriminator respectively.**

bodies and pending edges, leading to increased adhesion, as illustrated in the third column of Fig. 6. When both the generator and discriminator follow a "basic" structure, as seen in the red box in the second column of Fig. 6, the model struggles to differentiate the two foreground classes, resulting in erroneous enhancement. Abrupt texture and noise is obvious. Our designed discriminator focuses more on the semantic relationships among classes, enabling better differentiation. It also improves texture and shape in the enhanced image. With our discriminator, AJI, AJI+, Dice and PQ are increased by 0.3%, 1.7%, 1.1% and 3.1%. Compared to the basic generator, our generator improves all metrics to optimal levels, achieving scores of 0.721, 0.759, 0.860, 0.912, 0.853 and 0.778, respectively.

*5.3.2 Ablation Study on the Generator in Enhancement Model.* We demonstrate each component's role in the generator of the enhancement model. Integrating downsampling into the basic generator effectively aggregates nucleus pixel features, leading to a clearer separation between classes, as evidenced by the purple boxes in the fourth and fifth columns of Fig. 6. The improved differentiation directly reduces false positives and false negatives. Notably, Table 6 shows that SQ remains unchanged while DQ exhibits a 3.6% improvement, which validates our findings. Additionally, the introduction of the Res-CBAM module further suppresses irrelevant information and noise. Improvement in non-nucleus positions can be observed from the orange boxes in the fifth and sixth columns of Fig. 6. All evaluation metrics for this structure reach optimal levels.

## 6 CONCLUSION

In this study, we propose a general framework for nucleus segmentation in IHC images. It comprises two stages: initial enhancement followed by segmentation. A classic generative architecture and segmentation network demonstrate the effectiveness of our framework. In the enhancement model, we design a lightweight generator and discriminator to improve both enhancement robustness and computational efficiency. Comparisons against existing methods on both the private and public datasets highlight our method's SOTA accuracy. Cross-dataset evaluation further validates its superior generalization ability. Additionally, our method exhibits highly competitive processing speed. Further discussions on methodology and future work are presented in supplementary materials.

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
