# OpenReview forum: "GeNSeg-Net: A General Segmentation Framework for Any Nucleus in Immunohistochemistry Images"
_acmmm.org/ACMMM/2024/Conference — MM2024 Poster_

### Official Review · Reviewer_675d · 2024-05-14

**Rating:** 5
**Confidence:** 3

**Summary:**

The paper proposes a nuclei segmentation framework for IHC images called GeNSeg-Net. The framework includes an enhancement model, which is a GAN, and a segmentation model, which is a U-Net. The GAN model generates synthetic nuclei images from various types of IHC images. The U-Net model uses these synthetic nuclei labels to enhance its segmentation ability. The segmentation results show that the proposed framework surpasses some of the existing methods.

**Strengths:**

1. The framework is effective, efficient, and demonstrates strong cross-type generalization ability.
2. The paper is well-organized, and the experiments are sufficient.

**Limitations:**

1. In Figure 2, the detailed explanations of the 'Copy' and 'Transformation' between ideal nuclei and annotations are missing. Additionally, the usage of the enhanced image Y in U-Net is not clearly illustrated.

2. In contribution 1, what does 'training on a small subset of types' mean? Does this refer to semi-supervised learning or to generalization ability? What specialized model design have you implemented to achieve this?

3. Additional nuclei segmentation methods need to be compared to adequately demonstrate the effectiveness of the proposed method:
[1] Chen, Qi, et al. "PCTrans: Position-Guided Transformer with Query Contrast for Biological Instance Segmentation." Proceedings of the IEEE/CVF International Conference on Computer Vision. 2023.
[2] Hörst, Fabian, et al. "Cellvit: Vision transformers for precise cell segmentation and classification." Medical Image Analysis 94 (2024): 103143.

**Suitability:**

3

---

### Official Review · Reviewer_LLHe · 2024-05-23

**Rating:** 4
**Confidence:** 3

**Summary:**

This paper presents a nuclei segmentation framework by combining GAN and U-Net. The GAN model can enhance input images while the U-Net can use enhanced input to predict segmentation maps.

**Strengths:**

1) Using GAN to enhance image quality first is novel.

2) The proposed framework is simple and easy to catch. Experiments demonstrate the effectiveness of the proposed GeNSeg-Net.

**Limitations:**

1) There are some other nuclei datasets, such as MoNuSeg (A Multi-Organ Nucleus Segmentation Challenge), TNBC (Segmentation of Nuclei in Histopathology Images by Deep Regression of the Distance Map) and CPM-17 (Methods for segmentation and classification of digital microscopy tissue images). Are these images included in the DSB2018 and BBBC006v1 datasets? If not, it would be better to provide more experiment results.

2) Since this paper states the generator and discriminator are lightweight networks, could this paper provide a complexity analysis of GeNSeg-Net? For example, the params, FLOP and running time of GeNSeg-Net with and without this generator. How much did training costs increase by utilizing this GAN model?  How about the complexity of other compared methods?

3) Could this paper provide more details on how to calculate pixel-to-boundary Euclidean distance to further categorise each pixel and how to adjust their stain intensity?

4) As shown in Figure 2, the function of Generator is to generate initial segmentstion maps and followed by an U–Net to revise this segmentation maps. Based on this function, is this framework actually a two–step segmentation method?

5) Why this paper utilise a Generator in here instead of any segmentation networks?

6) What are the inputs of the U–Net?

**Suitability:**

3

---

### Official Review · Reviewer_Jf4Z · 2024-05-24

**Rating:** 4
**Confidence:** 3

**Summary:**

This paper introduces GeNSeg-Net, a novel framework for Immunohistochemistry (IHC) that effectively enhances and segments nuclei with minimal data, addressing the challenges of variability from several factors. It improves accuracy and generalization across different conditions, outperforming existing methods, and will be available for both research and clinical use.

**Strengths:**

1.High Generalization Ability: GeNSeg-Net can effectively segment nuclei across diverse tissue types and imaging techniques, even with high variability and limited training data. This broad applicability is essential for practical deployment in varied medical settings.
2.Enhanced Accuracy: The two-stage process involving an enhancement model and a segmentation model simplifies the segmentation task and leads to higher accuracy. This is achieved by standardizing nucleus morphology before segmentation.

**Limitations:**

1.Dependency on Specific Enhancements: The success of the segmentation relies heavily on the performance of the enhancement stage. Inaccuracies or failures in enhancement can propagate errors to the segmentation output.

**Suitability:**

2

---

### Meta-Review · Area_Chair_nAK6 · 2024-06-30

**Recommendation:** Accept (Poster)
**Confidence:** 5

**Metareview:**

This paper proposed a general segmentation for any nucleus in immunohistochemistry images. Three reviewers confirmed the merits of this paper and recommended acceptance with 1 weak accept and two borderline accepts. Given the consensus of reviewers, a decision of accept is recommended. I do suggest the authors revise the manuscript by taking all the reviewers’ comments into consideration.